# The Group Contribution to the Function Derived from Density and Speed-of-Sound Measurements for Glymes in *N*,*N*-Dimethylformamide + Water Mixtures

**DOI:** 10.3390/molecules28041519

**Published:** 2023-02-04

**Authors:** Małgorzata Jóźwiak, Marlena Komudzińska, Magdalena Tyczyńska

**Affiliations:** Department of Physical Chemistry, Faculty of Chemistry, University of Lodz, Pomorska 165, 90-236 Lodz, Poland

**Keywords:** pentaglyme, hexaglyme, density, speed of sound, *N*,*N*-dimethylforamide + water mixtures, hydrophobic hydration, –CH_2_– and –O– group contribution

## Abstract

The density and speed of sound of pentaglyme and hexaglyme in the *N*,*N*-dimethylformamide + water mixture at four temperatures are presented. The limiting apparent molar volumes ($V_{\Phi ,m}^{0}=V_{m}^{0}$), the isobaric molar thermal expansion ($E_{p,m}^{0}$), the isentropic compressibility ($\kappa_{S}^{}$), and the limiting partial molar isentropic compression ($K_{S,\Phi ,m}^{0}$ = $K_{S,m}^{0}$) were calculated. Changes in the values obtained from the physicochemical parameters, as functions of composition and temperature, were analyzed in terms of the molecular interactions and structural differentiation of the investigated systems. The hydrophobic hydration process of the studied glymes was visible in the area of high water content in the mixture. The hydration number of glymes in water at four temperatures was calculated and analyzed. The contribution of the –CH_2_– and –O– group to the functions describing the volume and acoustic properties of the investigated system was calculated. The calculated values of the functions analyzed using the group contribution are in agreement with the values obtained from the experimental data. Thus, such contributions are valuable for wide ranges of data, which can be used to analyze the hydrophobic hydration and preferential solvation processes, as well as to calculate the values of these functions for other similar compounds.

## 1. Introduction

The combination of results of densimetric and sonochemical studies enables a more in-depth description of the interactions between molecules in solutions. It provides valuable information in predicting the types of interactions that prevail in liquid mixtures, allowing us to draw conclusions about the solvation process, including hydrophobic hydration. These techniques have a wide range of applications and acceptance in many fields. The speed of sound, along with the density, is useful to calculate some important volumetric, acoustic, and thermodynamic properties of the solutions, but doing so also makes the interpretations of both types of data more reliable in relation to each other [1,2,3,4,5]. Ultrasonic and densimetric studies of liquid mixtures are of high practical importance in industry, in the control of manufacturing processes, and in chemical processing [6]. These methods are highly sensitive to molecular interactions and are particularly useful for elucidating solute–solute and solute–solvent interactions [7]. 

Aqueous–organic mixtures are an important group of solvents that are attracting increasing interest. Water–amide solutions are of practical importance in industry and in biological research [8]. *N*,*N*-dimethylformamide (DMF) is used as a solvent for gasses and liquids and in the production of acrylic fibers, vinyl polymers, films, coatings, and artificial leather [8]. DMF is associated with dipole–dipole interactions to some extent. Significant structural effects are absent due to the lack of hydrogen bonds [9]. Due to its miscibility with almost all common polar and nonpolar solvents [9,10,11,12] and because its hydrophilic–hydrophobic character is almost compensated, it is the best solvent in a mixture with water to analyze the solvation process, including hydrophobic hydration [13,14]. 

Glymes are compounds belonging to the group of polyethylene glycol methyl ethers with the general formula CH_3_O(CH_2_CH_2_O)_n_CH_3_. The number of oxyethylene groups (*n*) in a glyme molecule determines its polarity. With an increase in the number of the –OCH_2_CH_2_– groups, the hydrophobic nature of a linear polyether molecule increases [15]. Open chain polyethylene glycols have been found to be capable of forming complexes with different metal cations in organic solvents, in a similar manner to the macrocyclic polyethers [16,17]. Glymes have also been used as effective catalysts in phase transfer catalysis [18] and as effective solvents for many synthetic organic reactions [19]. Thus, open chain ethers and their derivatives are of increasing interest in many areas, because of their useful properties as effective complexing agents, as well as their relative inexpensiveness and nontoxicity [20]. It is worth paying attention to research on the behavior of glymes in a DMF and water (W) mixture.

In the present paper, we focus our attention on the solvation of selected glymes, CH_3_O(CH_2_CH_2_O)_n_CH_3_ where 5 ≤ *n* ≤ 6, in the DMF + W solvent using densimetry and sound of velocity methods. We suppose that hydrophobic hydration would manifest itself in water-rich solutions. Studies were carried out using both methods at four temperatures (293.15, 298.15, 303.15, 308.15) K in the entire range of the mixed solvent composition.

Furthermore, using the data resulting from the density and speed of sound propagation in the DMF + W mixtures containing glymes with shorter chains [21,22], the –CH_2_– and –O– groups’ contribution to the functions derived from the density and speed-of-sound measurements were calculated. 

## 2. Results and Discussion

### 2.1. Volumetric Properties

The apparent molar volume of the glymes ($V_{\Phi ,m}^{}$) values in the DMF + W mixture were calculated using the experimental values of the solution density via Equation (1).
(1)$$V_{\Phi ,m}^{}=\frac{M_{2}}{\rho }-\frac{1000\cdot \left(\rho -\rho_{0}\right)}{m\cdot \rho \cdot \rho_{0}}$$
where *M*_2_ is the molecular mass of glyme; $\rho ,\rho_{0}$ are the densities of solution and mixed solvent, respectively; and *m* is the concentration of glyme in moles of solute per kg of the mixed solvent. 

The obtained values are presented in Appendix A (see Appendix A).

Equation (2) (proposed by Redlich and Mayer) was used [23] to the analyze the function $V_{\Phi ,m}^{}=f(m)$.
(2)$$V_{\Phi ,m}^{}=V_{\Phi ,m}^{0}+b_{V}m$$
where $V_{\Phi ,m}^{0}$ is the limiting partial molar volume of the solute (glyme), *b*_V_ is the coefficient that provides information on the interaction between the solute (glyme) molecules, and *m* is the concentration of solution in moles of the solute (glyme) per kg of mixed solvent. 

The limiting apparent molar volume ($V_{\Phi ,m}^{0}$) for investigating the glymes in the mixed solvent was calculated using Equation (3) and is presented in Table 1 and Table 2.
(3)$$\underset{m\to 0}{\lim}V_{\Phi ,m}^{}=V_{\Phi ,m}^{0}=V_{m}^{0}$$
where $V_{m}^{0}$ is the limiting partial molar volume of glymes.

As seen in Table 1 and Table 2, the limiting partial molar volume of the glymes increased with the temperature increased. However, this increase was slight. It is known that an increase in temperature favors the weakening of the interactions between the molecules present in the solution, which causes an increase in the limiting partial molar volume of glymes in a mixture at a given composition.

Using the values of the limiting partial molar volume of glymes with shorter chains [21] in Figure 1, the function $V_{m}^{0}=f(x_{W})$ for glymes with 2 ≤ *m*_–O–_ ≤ 7 (*m*_–O–_ is the number of oxygen atoms in the glyme molecule) is presented.

As is seen in Figure 1, the curves are similar to each other for all the investigated glymes. They are typical for aqueous–organic systems and moreover illustrate the hydrophobic properties of the solute [21,24,25]. The limiting partial molar volume $V_{m}^{0}$ of the glymes increases with the increase in the number of oxyethylene groups in the glyme molecule (Figure 1). Additionally, with the increase in the number of oxyethylene groups in glyme molecules, their hydrophobic character increases. Thus, this arrangement of the curves on the graph is closely related to the information above. 

Using the values of the limiting partial molar volume of the glymes $V_{m}^{0}$ at four temperatures (*T*) and Equation (4), the isobaric molar thermal expansion of the glymes, $E_{p,m}^{0}$, in the DMF + W mixture was calculated, and the results are presented in Table 3 and in Figure 2 (together with analogous data for monoglyme, diglyme, triglyme, and tetraglyme [21]) as a function of *x*_W_ in the DMF + W mixed solvent.
(4)$$E_{p,m}^{0}={\left(\frac{\partial V_{m}^{0}}{\partial T}\right)}_{p}$$

As you can see in Figure 2, the shapes of all the presented curves are similar to each other. The course of the curve up to range *x*_W_ < 0.6 is gentle and then goes through a maximum at *x*_W_ ≈ 0.92. The intensity of the maximum increases with increasing hydrophobicity of the examined glymes. After passing through the maximum, there is a decrease in the value of limiting partial molar expansion $E_{p,m}^{0}$ of the glymes. This is related to the stiffening of the solvent structure caused by the hydrophobic hydration of the glymes. Within this range of mixed solvent composition, water is dominant in the mixture. The resulting phenomena are less susceptible to temperature changes. This effect increases with the increase in the hydrophobic nature of the glymes.

### 2.2. Acoustic Properties

The isentropic compressibility (*κ_S_*) values were calculated using Equation (5), and the obtained data are presented in Appendix A (see Appendix A).
(5)$$\kappa_{S}=\frac{1}{u^{2}\rho }$$
where *u* and *ρ* are the speed of sound and density of the glyme + DMF + W system, respectively. 

The values of the isentropic compressibility $\kappa_{S}$ of the isomolal glyme solution in the DMF + W mixed solvent, as a function of the mole fraction of water *x*_W_ in this mixture at 298.15 K, are presented in Figure 3 and Figure 4. Each of the lines represents the function $\kappa_{S}=f\left(x_{W}\right)$ at a given concentration of glyme, i.e., 0, 0.025, 0.05, 0.075, 0.125, 0.175, 0.2, 0.235, and 0.27. To make the intersection points of the curves more visible, the points that correspond to the values of the function have been omitted.

In Figure 3 and Figure 4, it can be seen that the $\kappa_{S}$ values reach a minimum at *x*_W_ ≈ 0.8 for all concentrations of the glyme in the DMF + W solvent. This decrease in the compressibility value is related to the incorporation of glyme molecules into the structure of the mixed solvent DMF + W. With a higher water content in the mixture *x*_W_ > 0.8, the isentropic compressibility values of the mixtures increase with increasing water content. This is probably related to the characteristic structure of the DMF + W mixture in this range of mixture composition. A similar situation was observed for the DMF + W systems containing shorter glyme chains [22].

Among others, de Visser et al. [26] believe that hydrogen bonds formed between water molecules and the carbonyl group of DMF molecules are stronger than hydrogen bonds formed between molecules in pure water. In addition, complexes may form in the DMF + W system when *x*_W_ < 0.8, which most likely causes a decrease in the compressibility coefficient of glyme solutions. In this region (*x*_W_ < 0.8), the glyme molecules are probably mainly solvated by DMF molecules. With the increase in water content, the DMF molecules present in the solvation shell of the glymes are exchanged for water molecules. Due to their properties, both glymes and DMF tend to stabilize the water structure. The presence of glyme molecules solvated by DMF and water molecules and the ordered structure of the mixed solvent most likely cause a decrease in the *κ*_S_ value in this composition range (*x*_W_ < 0.8).

The analysis of Figure 3 and Figure 4 shows the area where the isotherms of the isentropic compressibility versus the mole fraction of water show the intersection point before the function reaches the minimum value. Similar points are observed in the range of lower water content, i.e., *x*_W_ ≈ 0.15 and *x*_W_ ≈ 0.25, for pentaglyme and hexaglyme, respectively. The location of the intersection points is different for each glyme. According to Endo [27,28], in the compositions corresponding to the intersection points, clathrate structures may be formed in the range of rich water content in the mixture. In the resulting associations, the ratio of the number of solvent molecules to the number of solute molecules varies depending on the composition of the mixture. However, this interpretation is somewhat controversial and is discussed in the literature. It is probable that the dipole–dipole interactions of glyme molecules with DMF molecules (instead of interactions between glyme molecules and water molecules) cause the existence of second intersection points in the range of low water content in the mixture.

The location (*x*_W_) of the intersection points of isomolal functions $\kappa_{S}=f\left(x_{W}\right)$ for the investigated glymes, together with the literature values for glymes with shorter chains, is presented in Table 4.

As presented in Table 4, when the number of oxygen atoms in the glyme molecule increases, the water content in the mixture decreases for the first intersection point. For the second intersection points, the dependence is opposite.

In Figure 5 the dependence of the location (*x*_W_) of the first and second intersection points of the isomolal functions $\kappa_{S}=f\left(x_{W}\right)$ derived for glymes solutions in the mixture DMF + W as a function of the number of oxygen atoms in the glyme molecules (*m*_–O–_) is presented at different temperatures. In Figure 5a, the presented dependence is almost linear, but the function presented in Figure 5b is linear. The placement of the straight lines in Figure 5a is almost parallel, showing that the influence of the temperature on the first intersection point is similar for all glymes. In Figure 5b in the low water content of the mixture, the influence of diversified temperature is seen for long-chain glymes. One can notice that for hexaglyme, the position of the intersection point is almost not dependent on the temperature. Furthermore, according to Figure 5b, the less intensive temperature dependence for the second location (*x*_W_) of the intersection point is observed, i.e., the longer chain of the glyme. The mentioned influence of temperature is more visible in Figure 6a,b.

In Figure 6, the dependence of the composition of the solvent (*x*_W_) corresponding to the first and second intersection points of the isomolal curves $\kappa_{S}=f\left(x_{W}\right)$ in relation to the temperature obtained for the examined glymes in the DMF + W mixed solvent is presented. In this case, this dependence is seen to be linear. As the temperature increases, this intersection point shifts towards a lower water content in the mixture. In a high water content, this may indicate that at a higher temperature, there is a greater disturbance in the clathrate-like structure as a result of increased thermal movements. In the mixture where DMF is dominating as a solvent, the location (*x*_W_) of the second intersection point increases with the temperature increase. The parameters of the linear function $x_{W}=a+b\cdot T$ presented in Figure 6 are shown in Table 5. The determination coefficient and the standard deviation show that this linear relationship is very good. Using the values presented in Table 5, the location of the first and second intersection points can be calculated at a selected temperature. 

The values of apparent molar isentropic compression of glymes ($K_{S,\Phi ,m}^{}$) in the DMF + W solvent were calculated using Equation (6) and are presented in Appendix A (see Appendix A):(6)$$K_{S,\Phi ,m}^{}=\frac{M_{2}\kappa_{S}}{\rho }+\frac{\kappa_{S}\rho_{0}-\kappa_{S,0}\rho }{m\rho \rho_{0}}$$
where *M*_2_ is the molar mass of glyme; $\rho ,\rho_{0}$ are the densities of solution and mixed solvent, respectively; $\kappa_{S}$, $\kappa_{S,0}$ are the isentropic compressibility coefficients of the solution and solvent, respectively; and *m* is the concentration of the solution in moles of solute per kg of mixed solvent.

Equation (7) was used for analyzing the function $K_{S,\Phi ,m}^{}=f(m)$.
(7)$$K_{S,\Phi ,m}^{}=K_{S,\Phi ,m}^{0}+b_{K}m$$

The values of $K_{S,\Phi ,m}^{0}$ and *b*_K_ were obtained by using the least-squares method. The limiting apparent molar isentropic compression ($K_{S,\Phi ,m}^{0}$) is equal to the limiting partial molar isentropic compression of glyme ($K_{S,m}^{0}$). The values of $K_{S,m}^{0}$ for investigated glymes in the mixed solvent were calculated using Equation (8) and are presented in Table 6 and Table 7.
(8)$$\underset{m\to 0}{\lim}K_{S,\Phi ,m}^{}=K_{S,\Phi ,m}^{0}=K_{S,m}^{0}$$

As seen in Table 6 and Table 7, the standard partial molar compression of pentaglyme and hexaglyme in the DMF + W mixed solvent increases with increasing temperature, similar to that in the case of glymes with shorter chains [22]. 

As can be seen in Figure 7, the longer the chain of glymes and the greater the hydrophobic nature of these molecules, the greater the limiting molar value of isentropic compressibility of glymes. The limiting partial molar isentropic compression of tri-, tetra-, penta-, and hexaglyme passes through maxima in the range of medium water content in the mixture DMF + W, i.e., *x*_W_ < 0.7. The height of this maximum increases with the increase in the hydrophobic nature of the studied glymes. This may suggest that in the high and medium DMF content of the mixed solvent, we can observe weaker interaction between the glyme and mixed-solvent molecules. This may be the reason for the weakening of the bonds’ structure in the system and the contribution to a slightly greater compression of the system in this range of concentrations of the mixture. This conclusion is confirmed by the lack of observed changes in the limiting partial molar expansion of glymes in the same concentration range of the DMF + W mixture, as shown in Equation (4). For shorter chain glymes (monoglyme and diglyme), one can observe only the change in the slope of the curve.

In the range of high water content in the mixture (*x*_W_ > 0.7), the compression of the investigated systems decreases with increasing water content. The higher the water content in the mixture, the more clearly the process of the hydrophobic hydration of glymes, which have a hydrophobic character, is observed. This stiffens the structure of the system and thus reduces the compressibility [22,29].

In Figure 8, the $\frac{\partial K_{S,m}^{0}}{\partial T}$ values as a function of water content (*x*_W_) are presented for the glymes in the mixture DMF + W. As in the case of other functions, the greatest changes are seen with respect to the large amount of water in the system. The greatest influence of temperature on $K_{S,m}^{0}$ is observed for solutions with a high water content, in which the hydrophobic hydration of the glymes plays a significant role.

### 2.3. Hydration Number

Using the values of isentropic compressibility for the investigated glymes, the hydration numbers of the glymes ($n_{h}^{0}$) were calculated using a modified Pasynski’s method. This method is described, and its advantages and disadvantages are discussed in the literature by Burakowski and Gliński [30,31,32]. 

The hydration numbers of the investigated glyme molecules were calculated as described in our previous paper [22] and are presented in Table 8 and Figure 9 as a function of the number of oxygen atoms ($m_{-O-}$) in the glyme’s molecule.

As seen in Figure 9, the course of the function $n_{h}^{0}=f\left(m_{-O-}\right)$ is approximately linear at a constant temperature; i.e., the larger the glyme molecule, the more water molecules there are in the hydration shell. The dependence of the hydration numbers ($n_{h}^{0}$) on temperature is presented in Figure 10. 

It can be observed that this dependence is linear and can be described by Equation (9), and its parameters are presented in Table 9.
(9)$$n_{h}^{0}=c+d\cdot T$$

As seen in Figure 10 and Table 9 (b coefficient), the influence of temperature on the change in hydration number decreases with a decreasing number of oxygen atoms in the glyme molecules. This is probably related to the thermal movements of the chain in the glyme molecules. The longer the chain, the more degrees of freedom it has. From Figure 9, we can conclude that with the increase in molecule size, the bigger number of water molecules is probably moved from the shell with increasing temperature. 

### 2.4. The Contribution of the –CH_2_– and –O– Groups 

The values of limiting molar volume ($V_{m}^{0}$), limiting molar expansion ($E_{p,m}^{0}$), and limiting molar compression ($K_{S,m}^{0}$) for the solution of glymes with the number of oxygen atoms in molecule 2 ≤ *m*_–O–_ ≤ 7 made it possible to calculate the contribution of the –CH_2_– and –O– groups in these values. Using the analogous method proposed by Savage and Wood [33] the contribution of groups to the enthalpy of solvation of different compounds in pure solvent to be calculated. Thus, the average effect of the interaction between the –CH_2_– or –O– group (in glyme molecule) and molecules of the DMF + W mixed solvent for the values of $V_{m}^{0}$, $E_{p,m}^{0}$, and $K_{S,m}^{0}$ for glymes with 2 ≤ *m*_–O–_ ≤ 7 in the DMF + W mixture has been calculated.

For the calculation, it was assumed that –CH_3_ is equal to 1,5∙–CH_2_– [34], and Equation (10), as well as the nonlinear regression method, was used.
(10)Z=(3+2n)·P1+(1+n)·P2
where *Z* = $V_{m}^{0}$, $E_{p,m}^{0}$, and $K_{S,m}^{0}$; *n* is the number of the –CH_2_CH_2_O– group in the glyme molecule; *P*1 is the contribution of the –CH_2_– group; and *P*2 is the contribution of the ether oxygen atom –O–.

The obtained values of the average effect of the interaction between the –CH_2_– or –O– group and molecules of the DMF + W mixed solvent are presented in Table 10, Table 11 and Table 12 for $V_{m}^{0}$, $E_{p,m}^{0}$, and $K_{S,m}^{0}$, respectively, and in Figure 11. 

As can be seen in Figure 11, the courses of the functions describing the contribution delivered by the –CH_2_– group and the ether oxygen atom –O– to the $V_{m}^{0}$, $E_{p,m}^{0}$, and $K_{S,m}^{0}$ functions, depending on the mole fraction of water *x*_W_ in the mixed solvent, are opposite to each other. The contribution of the –CH_2_– group is positive, and, by the ether oxygen atom –O–, negative in all analyzed functions in the entire range of composition of the mixed solvent DMF + W.

In the range of high water content in the DMF + W mixed solvent characteristic, changes are observed in the functions $V_{m}^{0}=f\left(x_{2}\right)$ and $E_{m}^{0}=f\left(x_{2}\right)$, and these prove that the hydrophobic nature of the –CH_2_– group influences the structure of the mixture. Analogous opposite changes are observed in the case of the hydrophilic oxygen atom. Moreover, the contribution of the –CH_2_– and –O– to the function $E_{m}^{0}=f\left(x_{2}\right)$ is almost constant for small and medium water content in the mixture. In the case of group contributions obtained for standard partial molar compression, it can be seen that the function $K_{S,m}^{0}=f\left(x_{2}\right)$ curve is monotonic. No characteristic course of the functions is observed in the area with high water content. This is probably related to the lower sensitivity of this function to changes in the hydrophobic–hydrophilic character. The increasing contribution of the atom –O– to the $K_{S,m}^{0}$ suggests that the presence of an oxygen atom in the low and medium water content of the DMF + W mixture also contributes to increasing the values of $K_{S,m}^{0}$ in this range of the mixture. 

For example, in Figure 12, the function courses $V_{m}^{0}=f\left(x_{2}\right)$, $E_{m}^{0}=f\left(x_{2}\right)$, and $K_{S,m}^{0}=f\left(x_{2}\right)$, calculated using the group contribution and calculated using experimental data for tetraglyme at 298.15 K in the DMF + W mixed solvent, are compared. As shown in Figure 12, the consistency of the obtained dependencies is satisfactory.

## 3. Experimental

### 3.1. Materials

The compounds used in the study in this paper, as well as the suppliers, the purity, the purification method, and water content, are presented in Table 13. 

### 3.2. Method

The densities and speed of sound of the glyme + DMF + W solutions within the whole concentration range of the mixed solvent DMF + W at temperatures *T* = (293.15, 298.15, 303.15, 308.15) K were measured with the density and speed of sound analyzer DSA 5000 from Anton Paar. This device combines two miniaturized inline cells to simultaneously measure the density and speed of sound of a liquid sample at ambient pressure. The density is measured using a cell with oscillatory U-tube, where the repeatability of the density was ±1·10^–3^ kg·m^–3^, as declared by the manufacturer. Taking into account the formula for the combined standard uncertainty for the average of density measurements proposed by Fortin et al. [36], the estimated uncertainty according to this paper was ±4·10^−3^ kg·m^−3^. The sound-speed cell has a circular cylindrical cavity of 8 mm diameter and 5 mm thickness that is sandwiched between the transmitter and the receiver. The speed of sound is determined by measuring the time-of-flight of signals between the transmitter and receiver [36]. The repeatability declared by the manufacturer of the speed of sound measurements and their estimated uncertainty [36] were ±0.1 m·s^−1^ and ±0.5 m·s^−1^, respectively. Both measurement cells were housed in a thermostated block, the temperature of which was controlled with a combination of thermoelectric Peltier elements and an integrated Pt-100 resistance thermometer. The temperature measured with an integrated Pt-100 thermometer gives a repeatability in measured temperature equal to ±0.001 K. In the densimeter and sound-speed cells, an adjustment procedure was performed with ultra-pure Type 1 (MilliporeSigma™ Synergy™ Ultrapure Water Purification System), degassed water, and air at 293.15 K and at 0.1000 MPa pressure. The values of water density and speed of sound, amounting 998.203 kg·m^−3^ and 1482.66 m·s^−1^ at a temperature of 293.15 K, are similar to those reported in the literature [37].

The glymes solutions in the DMF + W mixture were prepared by weight using electronic balances with a precision of ±2·10^−5^ g. Temperature scans were programmed from 35 to 20 °C in decrements of 5 °C. The data of density and speed of sound obtained as a function of the water mole fraction (*x*_W_) of the solution and temperature *T* are presented in Appendix A (see Appendix A). 

## 4. Conclusions

The densimetric and acoustic investigations of glymes solutions in the DMF + W mixed solvent allow us to make a conclusion about the solvation process of glymes in this mixture at different temperatures. Analysis of the values of standard molar volume, standard molar thermal expansion, compressibility, and standard molar compression as a function of the mole fraction of water in the DMF + W mixed solvent ($V_{m}^{0}=f(x_{W})$, $E_{p,m}^{0}=f\left(x_{W}\right)$, $\kappa_{S}=f\left(x_{W}\right)$ and $K_{S,m}^{0}=f\left(x_{W}\right)$) shows the characteristic changes in these functions in the range of high water content in the mixture. These changes probably correlate with the hydrophobic hydration process of glyme molecules in a mixture with high water content.

Slight changes in these functions in the area of medium and low water content in the mixture show a stable structure of the mixed solvent in this range of compositions. This is consistent with the conclusions of other authors regarding the DMF + W solvent. 

The hydration number of glymes with a number of oxygen atoms in the chain 2 ≤ *m*_–O–_ ≤ 7 increases proportionally with the chain-length glymes and decreases linearly with increasing temperature.

The parameters of the linear relationship of the first and second intersection points of the isentropic compressibility $\kappa_{S}$ of isomolal solutions are presented in the current paper. Additionally, the hydration number as a function of the number of oxygen atoms in the glyme molecules, as well as the temperature, is presented ($x_{W}=f\left(m_{-O-}\right)$, $x_{W}=f\left(T\right)$, $n_{h}^{0}=f\left(m_{-O-}\right)$, $n_{h}^{0}=f\left(T\right)$). Knowing these parameters will allow one to calculate and interpret the intersection points of the isomolal curves of the isentropic compressibility and also calculate the hydration numbers for other similar systems. In addition, we hope that these relations will contribute in the future to improving the interpretation of the intersection points of the isentropic compressibility curves, as well as, contribute to finding an uncritical method for calculating hydration numbers. Of course, this also requires research on other systems, but our research will also contribute to this accordingly.

The calculated contribution of the –CH_2_– and –O– groups in the functions $V_{m}^{0}=f(x_{W})$, $E_{p,m}^{0}=f\left(x_{W}\right)$, and $K_{S,m}^{0}=f\left(x_{W}\right)$ is satisfactory and shows opposite changes in the course of these group contributions in the entire concentration rage of the DMF + W mixed solvent at 298.15 K.

## Figures and Tables

**Figure 1 molecules-28-01519-f001:**
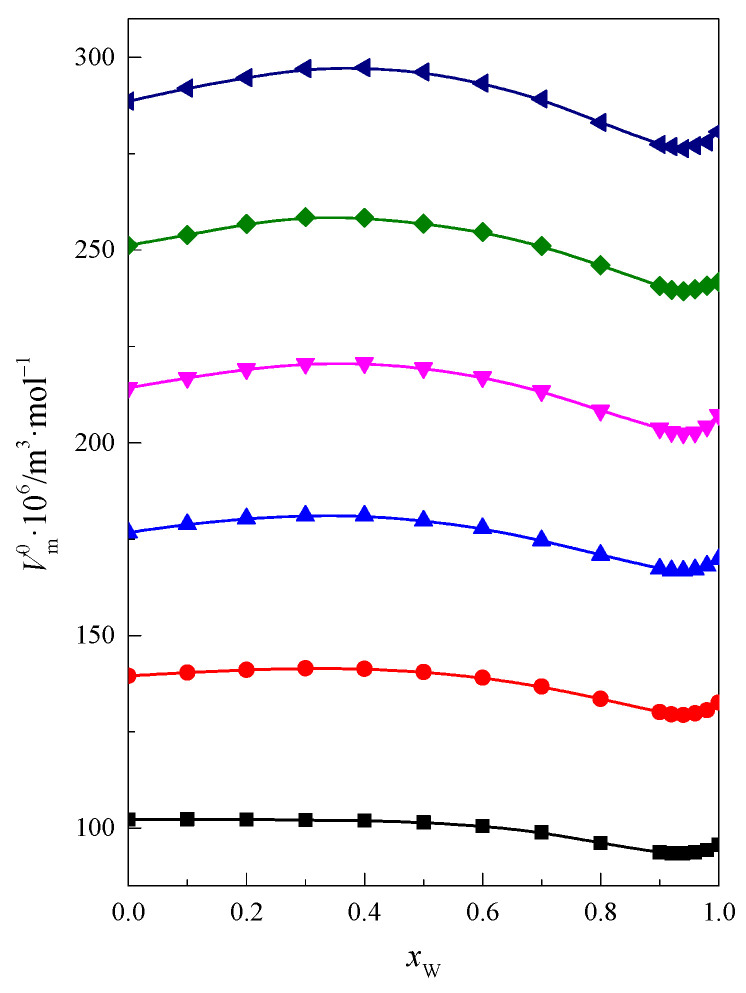
Limiting partial molar volume of solute in the DMF + W mixed solvent at 298.15 K: ■ monoglyme [21], ● diglyme [21], ▲ triglyme [21], ▼ tetraglyme [21], ◆ pentaglyme, ◀ hexaglyme.

**Figure 2 molecules-28-01519-f002:**
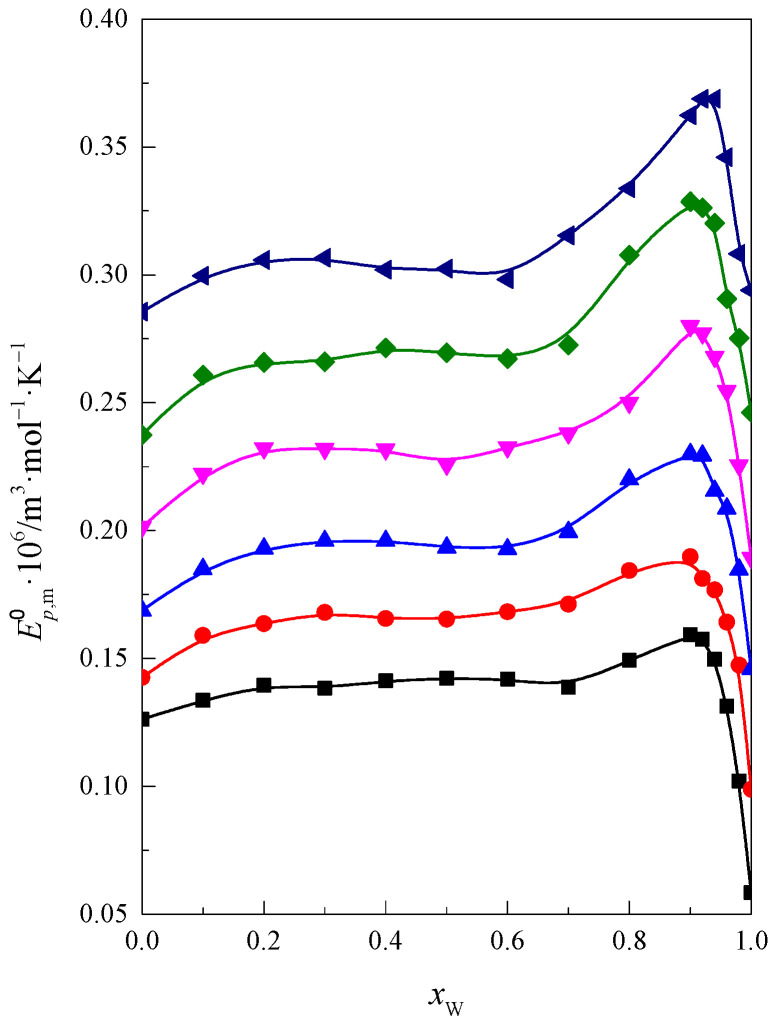
Standard isobaric molar thermal expansion of glymes in the DMF + W mixed solvent at 298.15 K: ■ monoglyme [21], ● diglyme [21], ▲ triglyme [21], ▼ tetraglyme [21], ◆ pentaglyme, ◀ hexaglyme.

**Figure 3 molecules-28-01519-f003:**
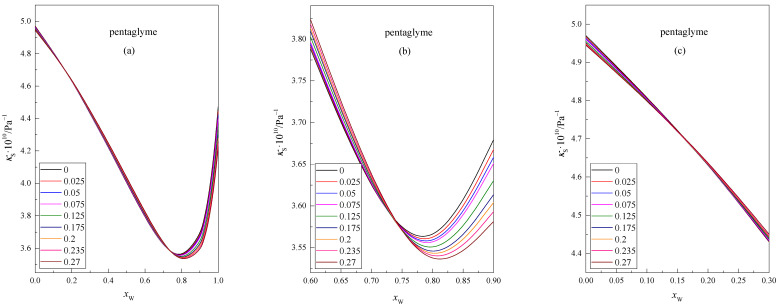
Dependence of the isentropic compressibility $\kappa_{S}$ of isomolal solutions of pentaglyme in the DMF + W mixed solvent in relation to the mole fraction of water, *x*_W_, at 298.15 K, (**a**) 0 ≤ *x*_W_ ≤ 1, (**b**) 0.6 ≤ *x*_W_ ≤ 0.9, (**c**) 0 ≤ *x*_W_ ≤ 0.3.

**Figure 4 molecules-28-01519-f004:**
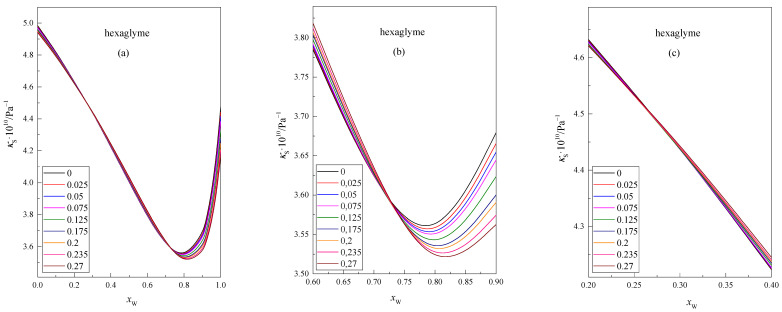
Dependence of the isentropic compressibility $\kappa_{S}$ of isomolal solutions of hexaglyme in the DMF + W mixed solvent in relation to the mole fraction of water, *x*_W_, at 298.15 K, (**a**) 0 ≤ *x*_W_ ≤ 1, (**b**) 0.6 ≤ *x*_W_ ≤ 0.9, (**c**) 0.2 ≤ *x*_W_ ≤ 0.4.

**Figure 5 molecules-28-01519-f005:**
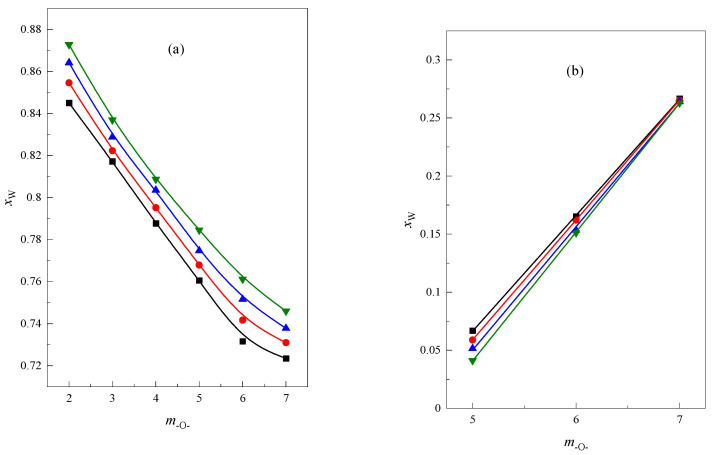
The location (*x*_W_) of the first (**a**) and second (**b**) intersection point of the isomolal functions $\kappa_{S}=f\left(x_{W}\right)$ derived for glyme solutions in the DMF + W mixed solvent as a function of the number of oxygen atoms in the glyme molecule (*m*_–O–_) at the temperatures of ■ 293.15 K, ● 298.15 K, ▲ 303.15 K, ▼ 308.15 K.

**Figure 6 molecules-28-01519-f006:**
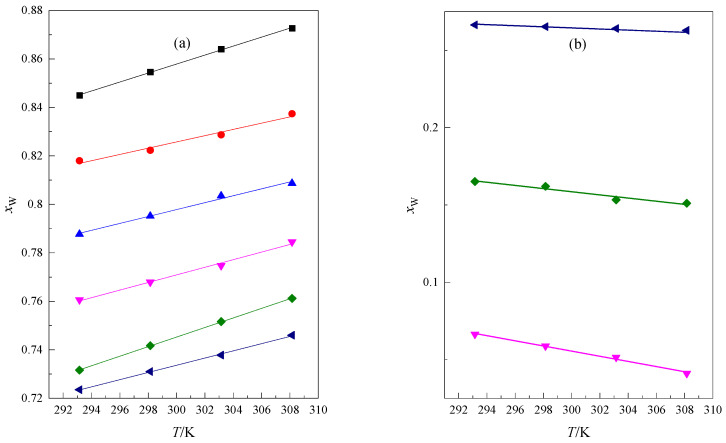
Composition of the solvent (*x*_W_) corresponding to the first (**a**) and second (**b**) intersection points of the isomolal curves $\kappa_{S}=f\left(x_{W}\right)$ as a function of temperature for the examined glymes in the DMF + W mixed solvent: ■ monoglyme [22], ● diglyme [22], ▲ triglyme [22], ▼ tetraglyme [22], ◆ pentaglyme, ◀ hexaglyme.

**Figure 7 molecules-28-01519-f007:**
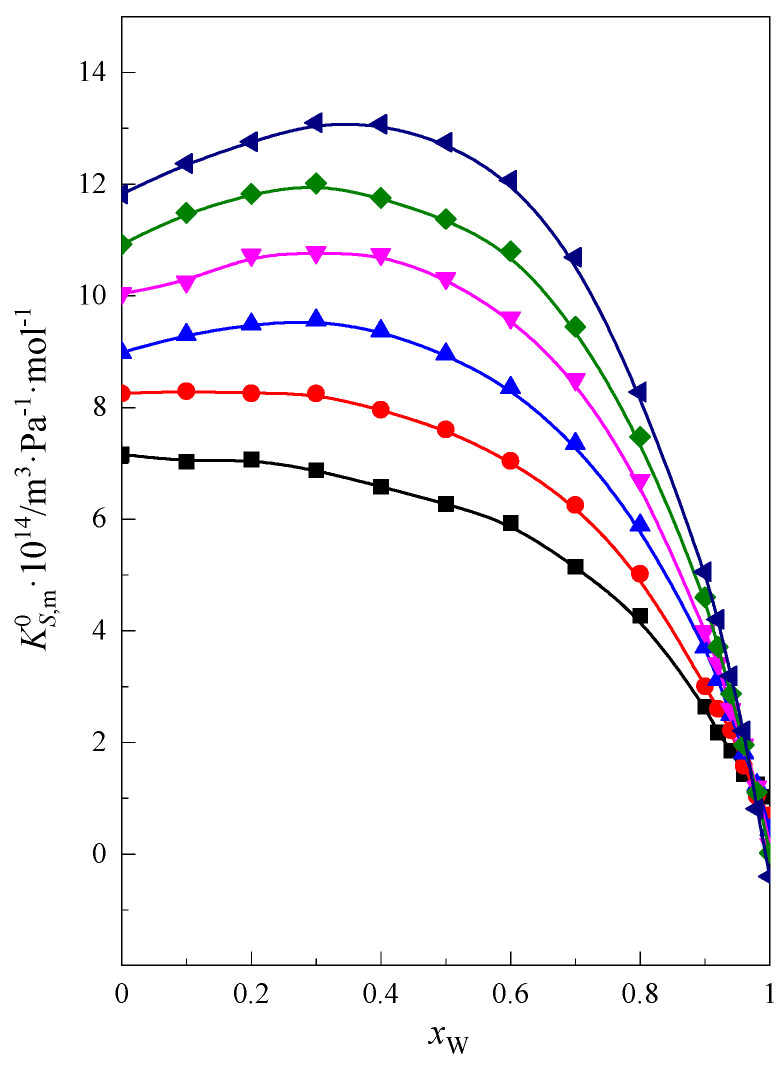
Standard partial molar compression of glymes in the DMF + W mixed solvent at a temperature of 298.15 K: ■ monoglyme [22], ● diglyme [22], ▲ triglyme [22], ▼ tetraglyme [22], ◆ pentaglyme, ◀ hexaglyme.

**Figure 8 molecules-28-01519-f008:**
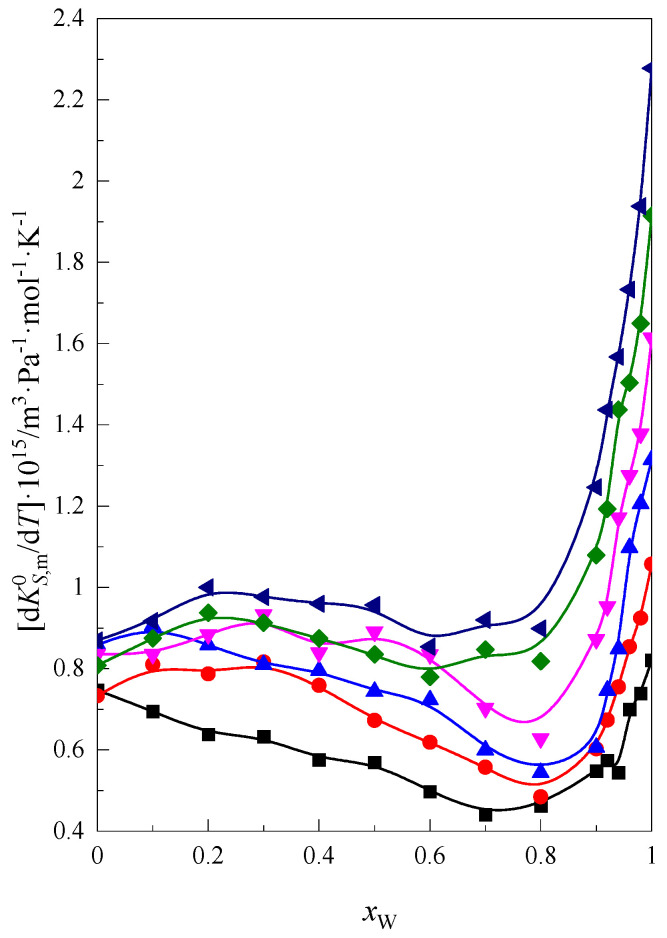
The $\frac{\partial K_{S,m}^{0}}{\partial T}$ as a function of water content (*x*_W_) in the mixture DMF + W: ■ monoglyme, ● diglyme, ▲ triglyme, ▼ tetraglyme, ◆ pentaglyme, ◀ hexaglyme.

**Figure 9 molecules-28-01519-f009:**
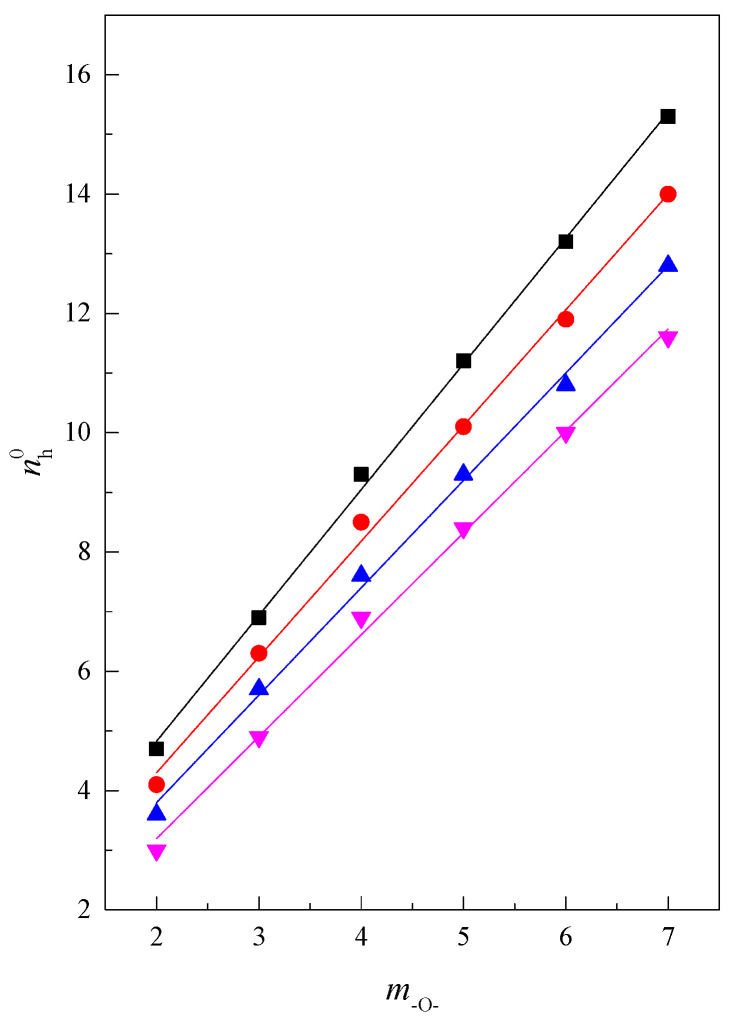
The hydration number ($n_{h}^{0}$) as a function of number of oxygen atoms (*m*_–O–_) in the glyme molecule at temperatures of ■ 293.15 K, ● 298.15 K, ▲ 303.15 K, ▼ 308.15 K.

**Figure 10 molecules-28-01519-f010:**
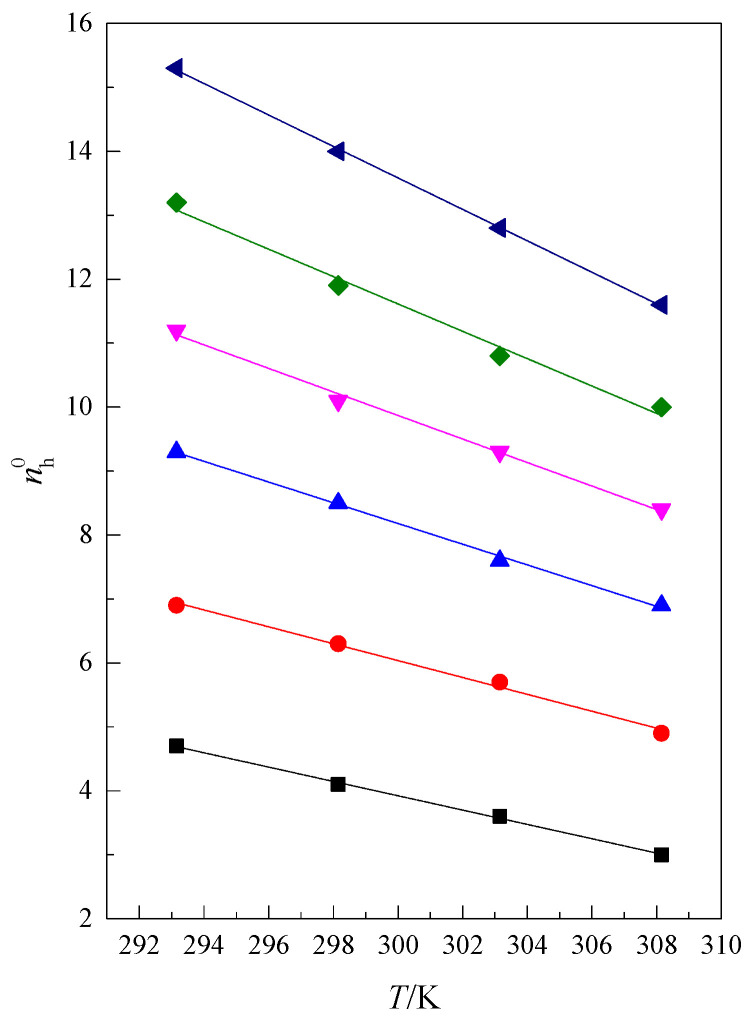
The hydration number ($n_{h}^{0}$) as a function of temperature (*T*) for ■ monoglyme, ● diglyme, ▲ triglyme, ▼ tetraglyme, ◆ pentaglyme, ◀ hexaglyme.

**Figure 11 molecules-28-01519-f011:**
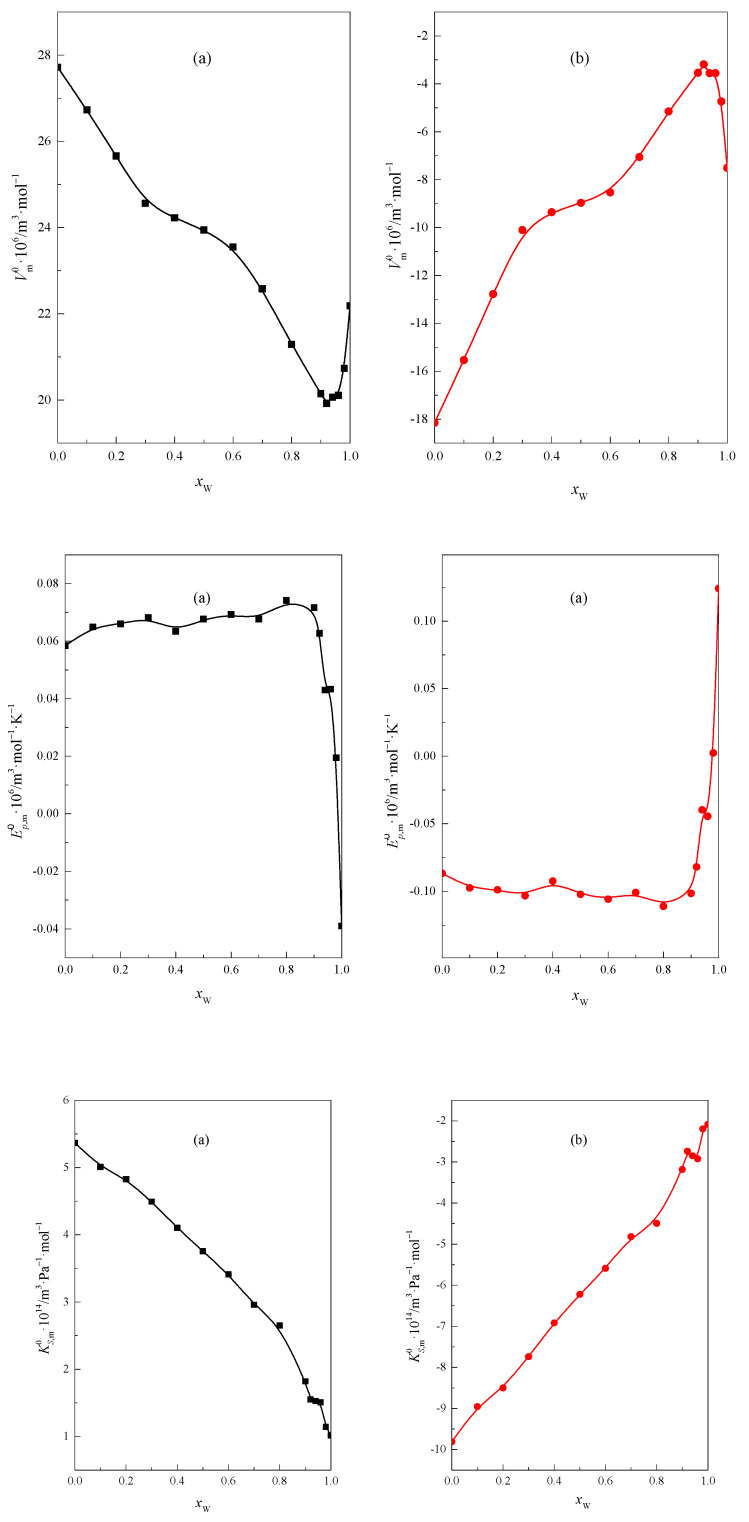
The average effect of interaction between the (**a**) –CH_2_– (■) or (**b**) –O– (●) group and molecules of the DMF + W mixed solvent for the values of $V_{m}^{0}$, $E_{p,m}^{0}$, and $K_{S,m}^{0}$ for glymes as a function of *x*_W_ at 298.15 K.

**Figure 12 molecules-28-01519-f012:**
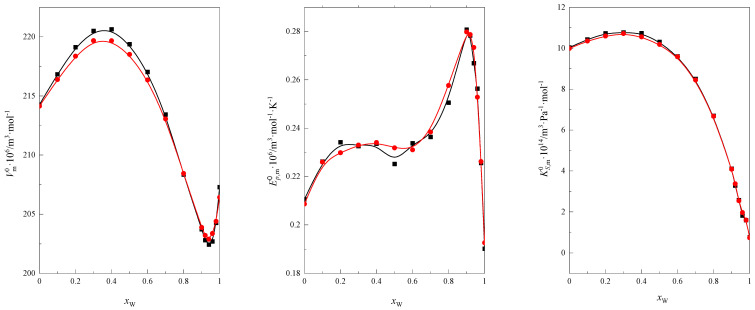
The courses of functions $V_{m}^{0}=f\left(x_{2}\right)$, $E_{m}^{0}=f\left(x_{2}\right)$, and $K_{S,m}^{0}=f\left(x_{2}\right)$ for tetraglyme in the DMF + W mixed solvent at 298.15 K have been calculated using experimental data [21,22], as represented by ■, and the groups’ contribution is represented by ●.

**Table 1 molecules-28-01519-t001:** Limiting partial molar volume of pentaglyme in the DMF + W mixed solvent.

*x* _W_	$V_{m}^{0}\cdot 10^{6}/m^{3}\cdot {\mathrm{mol}}^{-1}$			
	293.15 K	298.15 K	303.15 K	308.15 K
1.000	240.54 ± 0.01	241.76 ± 0.01	243.03 ± 0.01	244.22 ± 0.01
0.980	239.44 ± 0.01	240.86 ± 0.01	242.20 ± 0.01	243.58 ± 0.01
0.960	238.48 ± 0.01	239.91 ± 0.01	241.33 ± 0.01	242.85 ± 0.01
0.940	237.83 ± 0.01	239.41 ± 0.01	240.99 ± 0.01	242.64 ± 0.01
0.920	238.08 ± 0.01	239.64 ± 0.01	241.31 ± 0.01	242.96 ± 0.01
0.900	238.93 ± 0.01	240.66 ± 0.01	242.27 ± 0.01	243.87 ± 0.01
0.800	244.53 ± 0.01	246.08 ± 0.01	247.61 ± 0.01	249.15 ± 0.01
0.700	249.69 ± 0.01	251.03 ± 0.01	252.39 ± 0.01	253.78 ± 0.01
0.600	253.30 ± 0.01	254.63 ± 0.01	255.93 ± 0.01	257.32 ± 0.01
0.500	255.72 ± 0.01	257.06 ± 0.01	258.39 ± 0.01	259.77 ± 0.01
0.400	257.02 ± 0.01	258.42 ± 0.01	259.78 ± 0.01	261.09 ± 0.01
0.300	257.25 ± 0.01	258.52 ± 0.01	259.88 ± 0.01	261.23 ± 0.01
0.200	255.54 ± 0.01	256.90 ± 0.01	258.25 ± 0.01	259.53 ± 0.01
0.100	252.74 ± 0.01	254.05 ± 0.01	255.33 ± 0.01	256.66 ± 0.01
0.000	250.11 ± 0.02	251.37 ± 0.02	252.59 ± 0.02	253.66 ± 0.02

The uncertainty of the mole fraction *x*_W_ is equal to ±1·10^−3^.

**Table 2 molecules-28-01519-t002:** Limiting partial molar volume of hexaglyme in the DMF + W mixed solvent.

*x* _W_	$V_{m}^{0}\cdot 10^{6}/m^{3}\cdot {\mathrm{mol}}^{-1}$			
	293.15 K	298.15 K	303.15 K	308.15 K
1.000	279.24 ± 0.01	280.66 ± 0.02	282.16 ± 0.01	283.64 ± 0.01
0.980	276.42 ± 0.01	277.99 ± 0.01	279.51 ± 0.01	281.05 ± 0.01
0.960	275.31 ± 0.01	277.06 ± 0.01	278.82 ± 0.01	280.49 ± 0.01
0.940	274.40 ± 0.01	276.23 ± 0.01	278.11 ± 0.01	279.92 ± 0.01
0.920	274.88 ± 0.01	276.81 ± 0.01	278.60 ± 0.01	280.43 ± 0.01
0.900	275.54 ± 0.01	277.36 ± 0.01	279.19 ± 0.01	280.97 ± 0.01
0.800	281.43 ± 0.01	283.02 ± 0.01	284.65 ± 0.01	286.45 ± 0.01
0.700	287.60 ± 0.01	289.17 ± 0.01	290.78 ± 0.01	292.32 ± 0.01
0.600	291.78 ± 0.01	293.25 ± 0.01	294.75 ± 0.01	296.25 ± 0.01
0.500	294.73 ± 0.01	296.22 ± 0.01	297.73 ± 0.01	299.26 ± 0.01
0.400	295.82 ± 0.01	297.38 ± 0.01	298.83 ± 0.01	300.38 ± 0.01
0.300	295.49 ± 0.01	296.98 ± 0.01	298.54 ± 0.01	300.08 ± 0.01
0.200	293.26 ± 0.01	294.73 ± 0.01	296.28 ± 0.01	297.84 ± 0.01
0.100	290.49 ± 0.02	291.88 ± 0.02	293.48 ± 0.02	294.95 ± 0.02
0.000	287.24 ± 0.03	288.68 ± 0.03	290.09 ± 0.03	291.53 ± 0.03

The uncertainty of the mole fraction *x*_W_ is equal to ±1·10^−3^.

**Table 3 molecules-28-01519-t003:** Standard isobaric molar thermal expansion of glymes in the DMF + W mixture.

*x* _W_	$E_{p,m}^{0}\cdot 10^{6}/m^{3}\cdot {\mathrm{mol}}^{-1}\cdot K^{-1}$	
	Pentaglyme	Hexaglyme
1.000	0.246 ± 0.002	0.294 ± 0.002
0.980	0.275 ± 0.002	0.308 ± 0.001
0.960	0.291 ± 0.003	0.346 ± 0.003
0.940	0.320 ± 0.002	0.369 ± 0.002
0.920	0.326 ± 0.003	0.369 ± 0.004
0.900	0.329 ± 0.004	0.362 ± 0.002
0.800	0.308 ± 0.001	0.334 ± 0.007
0.700	0.273 ± 0.002	0.315 ± 0.002
0.600	0.267 ± 0.003	0.298 ± 0.001
0.500	0.270 ± 0.002	0.302 ± 0.001
0.400	0.271 ± 0.003	0.302 ± 0.003
0.300	0.266 ± 0.003	0.307 ± 0.002
0.200	0.266 ± 0.003	0.306 ± 0.003
0.100	0.261 ± 0.001	0.300 ± 0.005
0.000	0.237 ± 0.006	0.286 ± 0.001

The uncertainty of the mole fraction *x*_W_ is equal to ±1·10^−3^.

**Table 4 molecules-28-01519-t004:** The mole fraction of water (*x*_W_) in the DMF + W mixed solvent corresponding to the intersection point of the functions $\kappa_{S}=f\left(x_{W}\right)$ for the glymes at 298.15 K.

	*x* _W_			
	293.15 K	298.15 K	303.15 K	308.15 K
	the first intersection point			
monoglyme ^a^	0.844	0.854	0.864	0.875
diglymea	0.817	0.824	0.830	0.838
triglyme ^a^	0.788	0.795	0.802	0.809
tetraglyme ^a^	0.761	0.768	0.776	0.785
pentaglyme	0.733	0.743	0.752	0.762
hexaglyme	0.724	0.732	0.739	0.746
	the second intersection point			
tetraglyme ^a^	0.066	0.059	0.052	0.041
pentaglyme	0.165	0.162	0.153	0.151
hexaglyme	0.267	0.265	0.264	0.263

^a^ data have been taken from Ref. [22].

**Table 5 molecules-28-01519-t005:** The coefficients of the straight line equation $x_{W}=a+b\cdot T$ that describe the dependence of the mole fraction of water (*x*_W_) corresponding to the first and second intersection points in relation to temperature.

	a	b	*R* ^2^	*SD*
monoglyme	0.240 ± 0.0104	0.00206 ± 0.00003	0.99943	0.00038
diglyme	0.412 ± 0.016	0.00138 ± 0.00005	0.99707	0.00060
triglyme	0.378 ± 0.001	0.00140 ± 0.00001	0.99999	0.00001
tetraglyme	0.292 ± 0.019	0.00160 ± 0.00001	0.99688	0.00071
pentaglyme	0.170 ± 0.009	0.00192 ± 0.00003	0.99957	0.00032
hexaglyme	0.296 ± 0.001	0.00146 ± 0.00003	0.99888	0.00038
tetraglyme	0.556 ± 0.031	−0.00167 ± 0.00010	0.99234	0.00116
pentaglyme	0.465 ± 0.005	−0.00102 ± 0.00018	0.94385	0.00197
hexaglyme	0.336 ± 0.001	−0.00024 ± 0.00001	0.99993	0.00002

*R*^2^ is the regression coefficient. *SD* is the standard deviation.

**Table 6 molecules-28-01519-t006:** Standard partial molar compression of pentaglyme in the DMF + W mixed solvent.

*x* _W_	$K_{S,m}^{0}\cdot 10^{14}/\left(m^{3}\cdot {\mathrm{Pa}}^{-1}\cdot {\mathrm{mol}}^{-1}\right)$			
	293.15 K	298.15 K	303.15 K	308.15 K
1.000	−0.43 ± 0.02	0.69 ± 0.02	1.64 ± 0.01	2.44 ± 0.01
0.980	0.84 ± 0.04	1.71 ± 0.02	2.56 ± 0.03	3.31 ± 0.02
0.960	1.15 ± 0.02	2.02 ± 0.02	2.64 ± 0.02	3.45 ± 0.01
0.940	2.18 ± 0.01	2.68 ± 0.01	3.51 ± 0.02	4.30 ± 0.02
0.920	3.09 ± 0.01	3.69 ± 0.01	4.37 ± 0.01	4.86 ± 0.01
0.900	4.05 ± 0.01	4.62 ± 0.01	5.11 ± 0.01	5.69 ± 0.01
0.800	7.08 ± 0.01	7.45 ± 0.01	7.91 ± 0.02	8.29 ± 0.03
0.700	9.02 ± 0.01	9.44 ± 0.01	9.89 ± 0.02	10.28 ± 0.02
0.600	10.40 ± 0.02	10.80 ± 0.01	11.16 ± 0.01	11.58 ± 0.01
0.500	11.05 ± 0.02	11.44 ± 0.02	11.86 ± 0.02	12.30 ± 0.02
0.400	11.32 ± 0.02	11.73 ± 0.02	12.15 ± 0.02	12.64 ± 0.04
0.300	11.58 ± 0.01	12.00 ± 0.01	12.45 ± 0.01	12.95 ± 0.02
0.200	11.37 ± 0.02	11.84 ± 0.02	12.31 ± 0.01	12.78 ± 0.01
0.100	11.06 ± 0.02	11.49 ± 0.02	11.92 ± 0.02	12.37 ± 0.02
0.000	10.53 ± 0.02	10.92 ± 0.02	11.32 ± 0.02	11.74 ± 0.02

The uncertainty of the mole fraction *x*_W_ is equal to ±1·10^−3^.

**Table 7 molecules-28-01519-t007:** Standard partial molar compression of hexaglyme in the DMF + W mixed solvent.

*x* _W_	$K_{S,m}^{0}\cdot 10^{14}/\left(m^{3}\cdot {\mathrm{Pa}}^{-1}\cdot {\mathrm{mol}}^{-1}\right)$			
	293.15 K	298.15 K	303.15 K	308.15 K
1.000	–0.61 ± 0.02	0.64 ± 0.02	1.80 ± 0.02	2.80 ± 0.02
0.980	0.81 ± 0.02	1.77 ± 0.01	2.81 ± 0.01	3.70 ± 0.02
0.960	1.19 ± 0.01	2.25 ± 0.01	3.01 ± 0.02	3.82 ± 0.02
0.940	2.21 ± 0.02	2.93 ± 0.02	3.72 ± 0.01	4.56 ± 0.01
0.920	3.45 ± 0.02	4.16 ± 0.02	4.89 ± 0.01	5.61 ± 0.01
0.900	4.23 ± 0.01	4.88 ± 0.01	5.48 ± 0.01	6.10 ± 0.02
0.800	7.79 ± 0.01	8.28 ± 0.01	8.71 ± 0.01	9.15 ± 0.01
0.700	10.20 ± 0.02	10.69 ± 0.02	11.18 ± 0.01	11.57 ± 0.01
0.600	11.64 ± 0.02	12.07 ± 0.01	12.47 ± 0.01	12.93 ± 0.02
0.500	12.29 ± 0.01	12.75 ± 0.02	13.24 ± 0.02	13.72 ± 0.02
0.400	12.61 ± 0.01	13.07 ± 0.02	13.55 ± 0.02	14.05 ± 0.02
0.300	12.63 ± 0.02	13.10 ± 0.02	13.59 ± 0.02	14.09 ± 0.02
0.200	12.27 ± 0.01	12.76 ± 0.02	13.26 ± 0.02	13.78 ± 0.02
0.100	11.95 ± 0.02	12.37 ± 0.02	12.85 ± 0.02	13.32 ± 0.02
0.000	11.40 ± 0.02	11.82 ± 0.02	12.23 ± 0.01	12.71 ± 0.01

The uncertainty of the mole fraction *x*_W_ is equal to ±1·10^−3^.

**Table 8 molecules-28-01519-t008:** The hydration number of glymes.

Glyme	293.15 K	298.15 K	303.15 K	308.15 K
monoglyme ^a^	4.7	4.1	3.6	3.0
diglyme ^a^	6.9	6.3	5.7	4.9
triglyme _a_	9.3	8.5	7.6	6.9
tetraglyme ^a^	11.2	10.1	9.3	8.4
pentaglyme	13.2	11.9	10.8	10.0
hexaglyme	15.3	14.0	12.8	11.6

^a^ data were taken from Ref. [22].

**Table 9 molecules-28-01519-t009:** The parameters of Equation (9).

	c	d	*R* ^2^	*SD*
monoglyme ^a^	37.5 ± 0.9	−0.112 ± 0.003	0.99879	0.032
diglyme ^a^	45.6 ± 2.1	−0.132 ± 0.007	0.99452	0.077
triglyme ^a^	56. 8 ± 1.6	−0.162 ± 0.005	0.99787	0.059
tetraglyme ^a^	65.1 ± 2.6	−0.184 ± 0.008	0.99576	0.095
pentaglyme	75.8 ± 4.8	−0.214 ± 0.016	0.98911	0.178
hexaglyme	87.4 ± 1.0	−0.246 ± 0.003	0.99360	0.039

^a^ data were taken from Ref. [22]. *R*^2^ is the regression coefficient. *SD* is the standard deviation.

**Table 10 molecules-28-01519-t010:** The average effect of the interaction between the –CH_2_– or –O– group and molecules of the DMF + W mixed solvent in $V_{m}^{0}$ at 298.15 K.

	$V_{m}^{0}\cdot 10^{6}/m^{3}\cdot {\mathrm{mol}}^{-1}$			
*x* _W_	–CH_2_–	–O–	*R* ^2^	SD
1.000	22.21 ± 1.09	−7.58 ± 2.39	0.99981	0.946
0.980	20.80 ± 0.39	−4.87 ± 0.86	0.99998	0.335
0.960	20.17 ± 0.58	−3.70 ± 1.28	0.99996	0.505
0.940	20.13 ± 0.53	−3.71 ± 1.16	0.99996	0.458
0.920	20.04 ± 0.44	−3.43 ± 0.97	0.99998	0.383
0.900	20.14 ± 0.18	−3.53 ± 0.40	1.00000	0.157
0.800	21.58 ± 0.21	−5.78 ± 0.47	0.99999	0.186
0.700	22.69 ± 0.31	−7.30 ± 0.67	0.99999	0.266
0.600	23.53 ± 0.45	−8.48 ± 0.99	0.99998	0.393
0.500	24.03 ± 0.62	−9.16 ± 1.36	0.99996	0.537
0.400	24.47 ± 0.73	−9.89 ± 1.61	0.99994	0.638
0.300	24.97 ± 0.66	−10.99 ± 1.44	0.99995	0.569
0.200	26.04 ± 0.66	−13.61 ± 1.45	0.99995	0.575
0.100	27.17 ± 0.41	−16.49 ± 0.91	0.99998	0.359
0.000	28.17 ± 0.26	−19.14 ± 0.56	0.99999	0.223

*R*^2^ is the regression coefficient. *SD* is the standard deviation. The uncertainty of the mole fraction *x*_W_ is equal to ±1·10^−3^.

**Table 11 molecules-28-01519-t011:** The average effect of interaction between the –CH_2_– or –O– group and molecules of the DMF + W mixed solvent in $E_{p,m}^{0}$ at 298.15 K.

	$E_{p,m}^{0}\cdot 10^{6}/m^{3}\cdot {\mathrm{mol}}^{-1}\cdot K^{-1}$			
*x* _W_	–CH_2_–	–O–	*R* ^2^	SD
1.000	–0.031 ± 0.008	0.107 ± 0.018	0.99442	0.007
0.980	0.020 ± 0.002	0.001 ± 0.004	0.99972	0.001
0.960	0.041 ± 0.006	−0.039 ± 0.012	0.99694	0.005
0.940	0.038 ± 0.011	−0.029 ± 0.024	0.99109	0.009
0.920	0.056 ± 0.006	−0.068 ± 0.014	0.99643	0.006
0.900	0.071 ± 0.006	−0.099 ± 0.012	0.99687	0.005
0.800	0.064 ± 0.006	−0.089 ± 0.014	0.99553	0.005
0.700	0.062 ± 0.006	−0.089 ± 0.013	0.99508	0.005
0.600	0.056 ± 0.006	−0.076 ± 0.013	0.99497	0.005
0.500	0.060 ± 0.006	−0.085 ± 0.012	0.99552	0.005
0.400	0.057 ± 0.003	−0.078 ± 0.006	0.99888	0.002
0.300	0.062 ± 0.003	−0.090 ± 0.006	0.99895	0.002
0.200	0.059 ± 0.004	−0.083 ± 0.008	0.99790	0.003
0.100	0.062 ± 0.008	−0.091 ± 0.017	0.99077	0.007
0.000	0.058 ± 0.004	−0.087 ± 0.009	0.99664	0.004

*R*^2^ is the regression coefficient. *SD* is the standard deviation. The uncertainty of the mole fraction *x*_W_ is equal to ±1·10^−3^.

**Table 12 molecules-28-01519-t012:** The average effect of interaction between the –CH_2_– or –O– group and molecules of the DMF + W mixed solvent in $K_{S,m}^{0}$ at 298.15 K.

	$K_{S,m}^{0}\cdot 10^{14}/m^{3}\cdot {\mathrm{Pa}}^{-1}\cdot {\mathrm{mol}}^{-1}$			
*x* _W_	–CH_2_–	–O–	*R* ^2^	SD
1.000	1.02 ± 0.02	−2.09 ± 0.05	0.97083	0.020
0.980	1.14 ± 0.02	−2.20 ± 0.03	0.99534	0.013
0.960	1.51 ± 0.15	−2.92 ± 0.32	0.70058	0.126
0.940	1.53 ± 0.14	−2.85 ± 0.32	0.92040	0.126
0.920	1.55 ± 0.08	−2.74 ± 0.17	0.99266	0.065
0.900	1.82 ± 0.11	−3.19 ± 0.25	0.98949	0.098
0.800	2.65 ± 0.03	−4.49 ± 0.07	0.99970	0.029
0.700	2.96 ± 0.07	−4.82 ± 0.15	0.99929	0.061
0.600	3.41 ± 0.06	−5.59 ± 0.12	0.99965	0.049
0.500	3.76 ± 0.10	−6.22 ± 0.23	0.99890	0.089
0.400	4.10 ± 0.15	−6.92 ± 0.33	0.99767	0.131
0.300	4.49 ± 0.11	−7.74 ± 0.24	0.99866	0.096
0.200	4.83 ± 0.13	−8.50 ± 0.29	0.99780	0.113
0.100	5.01 ± 0.12	−8.95 ± 0.27	0.99771	0.107
0.000	5.37 ± 0.09	−9.81 ± 0.20	0.99841	0.077

*R*^2^ is the regression coefficient. *SD* is the standard deviation. The uncertainty of the mole fraction *x*_W_ is equal to ±1·10^−3^.

**Table 13 molecules-28-01519-t013:** Materials.

Name of Compound	Source	Purity	Purification Method	Mass Fraction of Water ^c^
pentaglyme	prepared as described in [15]	>0.99 ^a^		3·10^−3^
hexaglyme	prepared as described in [35]	>0.99 ^a^		3·10^−3^
*N*,*N-*dimethylformamide (DMF)	Sigma-Aldrich Poznan, Poland	0.99 ^b^	Distillation under reduced pressure	2·10^−4^

^a^ The purity of the compound has been determined by means of ^1^H NMR; see Appendix A. ^b^ Declared by the supplier. ^c^ Determined by the Karl Fischer method.

## Data Availability

Not applicable.

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
