# Peer review of "The Group Contribution to the Function Derived from Density and Speed-of-Sound Measurements for Glymes in N,N-Dimethylformamide + Water Mixtures"

_molecules, 2023, doi:10.3390/molecules28041519_

Round 1

Reviewer 1 Report

The manuscript presents the results of density and speed of sound measurements for glymes in water-DMF mixtures.  The authors do a good job of presenting the results of their measurements and data regression.

Here are my recommendations

1. The abstract would be greatly improved with a sentence discussing the value and utility of the study results.  (Why should a reader care about these results?)

2. The language throughout is poor with problems of grammar and vocabulary.  In many places the language obscures the meaning. The manuscript needs a thorough edit for language.

3.  (Page 2) Explain why increasing polarity makes a glyme molecule more hydrophobic.  That seems backwards.

4. (Page 3) Although the details of the measurements are reported in earlier publications, it would be helpful to give a brief (one or two paragraphs) overview of the methods in this paper.

5. Related to item 1, above.  The authors report a lot of data and regressed parameter values.  They have apparently done good and careful work. However, the value of these results to the reader is unclear.  Of what utility are these results to the reader?  What should a reader do with these results? Additionally, what phenomenological insights and understanding should the reader get from these results that could be more broadly applicable?  Please add some discussion on these topics.

Reviewer 2 Report

General remark:

In my opinion, the paper shows a refreshing analysis related to hydration processes For this  reason, I consider the paper can be accepted, and published in Molecules  after a minor corrections.Probably, if the authors included results of molecular dynamics, some of their conclusions could be verified.

Comment 1. Abstract

Please give at least one quantitative result.

Comment 2. Group contributions–CH2- and –O-

Even acoustic properties could be information about group contributions. It is necessary to support those conclusions with molecular dynamic simulations.

Comment 3. Materials

Authors say that pentaglyme and hexaglyme, were prepared by previous methodology. The 1H NMR must be included in support information with the purity analysis. Please, data of mono, di, tri and tetraglyme is missed.

Comment 4.  Molar fraction

Authors give molar fraction with two digits and with rounded number. Please give xW with the experimental value you got.

Comment 5.  Experimental pressure

Please give the uncertainty for each variable in tables table 2,3,4,7 and 8 the experimental pressure.

Comment 6.  Figure 3

I suggest that if you have a figure on it, it is possible to show the two different phenomena.

Comment 7.  Font size

Please check font size and journal style.
